# Machine Learning-Based Monitoring of DC-DC Converters in Photovoltaic Applications



Marco Bindi [1,*], Fabio Corti [2], Igor Aizenberg [3], Francesco Grasso [1], Gabriele Maria Lozito [1], Antonio Luchetta [1], Maria Cristina Piccirilli [1] and Alberto Reatti [1]

1    Department of Information Engineering, University of Florence, 50139 Firenze, Italy;
     francesco.grasso@unifi.it (F.G.); gabrielemaria.lozito@unifi.it (G.M.L.); antonio.luchetta@unifi.it (A.L.);
     mariacristina.piccirilli@unifi.it (M.C.P.); alberto.reatti@unifi.it (A.R.)
2    Department of Industrial Engineering, University of Perugia, 06125 Perugia, Italy; fabio.corti@unipg.it
3    Department of Computer Science, Manhattan College, Riverdale, NY 10471, USA;
     igor.aizenberg@manhattan.edu
*    Correspondence: m.bindi@unifi.it

**Abstract:** In this paper, a monitoring method for DC-DC converters in photovoltaic applications is presented. The primary goal is to prevent catastrophic failures by detecting malfunctioning conditions during the operation of the electrical system. The proposed prognostic procedure is based on machine learning techniques and focuses on the variations of passive components with respect to their nominal range. A theoretical study is proposed to choose the best measurements for the prognostic analysis and adapt the monitoring method to a photovoltaic system. In order to facilitate this study, a graphical assessment of testability is presented, and the effects of the variable solar irradiance on the selected measurements are also considered from a graphical point of view. The main technique presented in this paper to identify the malfunction conditions is based on a Multilayer neural network with Multi-Valued Neurons. The performances of this classifier applied on a Zeta converter are compared to those of a Support Vector Machine algorithm. The simulations carried out in the Simulink environment show a classification rate higher than 90%, and this means that the monitoring method allows the identification of problems in the initial phases, thus guaranteeing the possibility to change the work set-up and organize maintenance operations for DC-DC converters.

**Keywords:** DC-DC converters; prognostic analysis; multi-valued neuron neural network; support vector machine; Zeta converter

## 1. Introduction

The development of smart cities leads to an increase in the complexity of electrical grids, and new challenges need to be addressed, such as the spread of electric vehicles and the management of renewable energy systems [1,2]. In this sense, new devices, control techniques and monitoring methods are needed for proper energy management [3–7]. The technical optimization of the new electrical generators allows an increase in efficiency for renewable systems, but it is not sufficient for the correct distribution of this energy, which is difficult to predict and highly variable [8,9]. For this reason, the development of new algorithms capable of predicting production from renewable sources and managing electrical loads will be a very important field of interest for many researchers [10–12]. Furthermore, the development and control of devices used as an interface between generators and the grid, or generators and other devices, play a fundamental role in the correct distribution of energy [13–17]. In this sense, the control of DC-DC converters represents a very important aspect because they can be used as an interface with renewable energy systems producing a Direct Current (DC) and are essential for all those systems powered by batteries, such as electric vehicles. In addition to controlling these devices, it is very important to monitor their state of health during the operation of the electrical system.

In traditional diagnostic systems, the main objective is to identify and localize faults, which leads to a complete loss of functionality. On the other hand, in prognostic systems, the subject of the analysis is the malfunction condition. This means that the prognostic system focuses on slight variations from the nominal point of work, identifying partial losses of functionality that precede catastrophic failures. In this way, it is possible to organize maintenance operations, increasing the reliability of the electrical system and reducing recovery times.

In this paper, the definition of a prognostic analysis for DC-DC converters is carried out and verified through a simulation procedure in a Matlab-Simulink environment. The converter taken into consideration is a Zeta converter, which allows for a high voltage gain and low ripple in the output current using four passive components [18–20]. These components are the main subjects of the prognostic analysis, and their variations with respect to the nominal values are used as indexes of the state converter of health. In fact, when a malfunction occurs on a passive component, its value changes; this introduces a variation of the working point [21,22] and could produce catastrophic consequences. To make the simulations as close as possible to the real functioning of the converter, the parasites of the real active and passive components are considered in Simulink.

The specific case of prognostics addressed in this paper involves the DC-DC converter featuring a photovoltaic (PV) input. This introduces two additional challenges to the prognostic problem. The first is the non-linear current-voltage characteristic of the source, which can result in irregular trends (if compared with ideal voltage and current sources, often used in diagnostics and prognostics problems) of the converter current and voltages. The second is the functional relationship between the characteristics and the environmental quantities of temperature and irradiance. Both difficulties might lead to erroneous classifications of the working condition of the converter. To address this problem, a specific normalization approach is used to decouple the prognostic-sensitive quantities from the environmental-dependent nature of the source.

Prognosis is performed by means of a supervised machine-learning approach. Several sensitive electrical quantities are measured on the passive elements of the operating circuit in the time domain and are processed by a Multilayer neural network with Multi-Valued Neurons (MLMVN). This classifier falls in the category of supervised learning algorithm, and it presents three layers with complex weights. Thanks to its complex nature, the MLMVN is easily adaptable to the classification of electrical quantities, which are usually expressed by phasors. Compared to real feed-forward neural networks, this classifier presents a derivative-free learning algorithm that facilitates the correction of the weights and reduces the computational cost. In several applications, MLMVN offers a better generalization capability than other machine learning techniques and its implementation in power line monitoring is presented in [23], where frequency response measurements are processed. The performance of the MLMVN in this new application is compared to that obtained by using a Support Vector Machine (SVM), which is one of the most used techniques in the field of data classification [24,25].

The paper is organized as follows: Section 2 shows the main characteristics of the renewable energy system taken into consideration, the theoretical aspects of the prognostic procedure, and the use of the MLMVN, Section 3 presents the main results of the simulation procedure, Section 4 reports the result discussion, and Section 5 shows the conclusions.

## 2. Materials and Methods

The analysis method proposed focuses on a photovoltaic system constituted by a 230 W solar panel and a Zeta converter connected to a DC microgrid (48 V). The DC-DC converter must guarantee the energy transfer from the source to the grid, and several techniques can be used to achieve this goal, such as the Maximum Power Point Tracking (MPPT) control [26,27]. The MPPT algorithm's purpose is to control the converter duty-cycle (D) to ensure an optimal operating point is achieved on the PV source. If no condition is required on the output current and voltage, classic MPPT aims at setting the source

voltage as close as possible to the maximum power voltage. Since this voltage changes according to the environmental conditions, either a tracking algorithm (e.g., the Perturb & Observe) or a model-based algorithm (also based on machine-learning methods) should be used. In this paper, the MPPT algorithm is not simulated because it does not represent a fundamental aspect of the prognostic analysis. The main idea the monitoring procedure is based on is to fix the duty-cycle for the short time interval necessary to extract the voltage and current measurements. This avoids putting the converter out of service and allows the definition of its state of health without interrupting the energy transfer. Therefore, during the prognostic analysis, the duty-cycle of the converter is not varied to reach the maximum power point, thus limiting the variability of the measurements and facilitating the localization of malfunctions. Once the prognosis is finished, the MPPT algorithm can vary the duty-cycle again. Since the measurement procedure requires only a few periods at the converter switching frequency, the prognostic analysis does not significantly affect the energy production.

### 2.1. Photovoltaic Source

The energy source considered in this paper is a 230 W solar panel with 60 multicrystalline cells TW230P60-FA by Tianwei New Energy [28]. The main electrical characteristics of the panel are extracted from its datasheet and reported in Table 1, where $V_{MPP}$ and $I_{MPP}$ are the maximum power point voltage and current, respectively, $V_{OC}$ is the open-circuit voltage, and $I_{SC}$ is the short-circuit current.

**Table 1.** Characteristics of the photovoltaic panel at the Standard Test Condition.

| $V_{MPP}$ | $I_{MPP}$ | $V_{OC}$ | $I_{SC}$ | $\alpha_T$ | N Cell |
|---|---|---|---|---|---|
| 29.4 V | 7.82 A | 37.3 V | 8.22 A | 0.06%/°C | 60 |

Based on these characteristics, it is possible to implement an equivalent circuit model in a Simulink environment for the panel and extract the voltage–current curves as the solar irradiance and the working temperature change. Figure 1a,b shows these curves obtained for different values of irradiance and temperature.

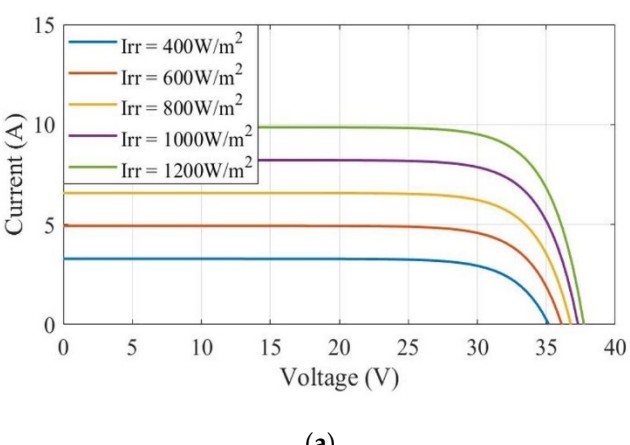
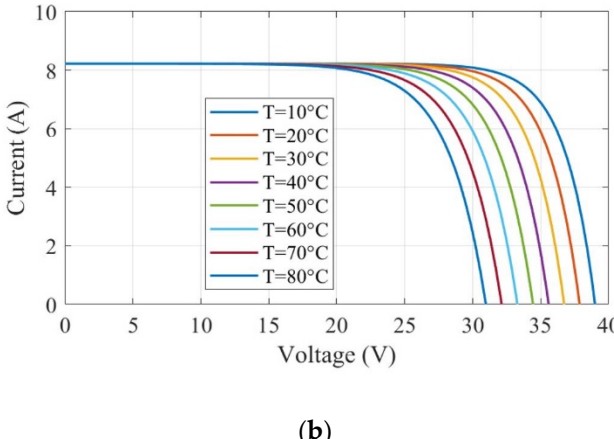

(**a**)　　　　　　　　　　　　　　　　　　　(**b**)

**Figure 1.** Voltage–Current curves of the photovoltaic panel; (**a**) curves obtained with fixed temperature (25 °C) as the irradiance varies, (**b**) curves obtained with fixed irradiance (1000 W/m$^2$) as the temperature varies.

Obviously, the input current and voltage depend on the environmental conditions, and this is reflected in the internal electrical quantities of the DC-DC converter. Since the measurements extracted from the DC-DC converter for evaluating its state of health are sensitive to the changes in the input current and voltage, those measurements are sensitive

to the environmental conditions of the PV device as well. This can create confusion during the classification of malfunctions because the monitoring system must be able to discriminate the variations introduced by the aging of the components from those caused by changes in solar irradiance and working temperature.

To avoid this problem, a simple solution could be to add the values of the irradiance and temperature to the set of measurements processed by the classifier. However, these quantities are not easily measurable from a practical point of view, and this solution makes the training stage more complex by requiring a very large dataset. In this paper, a graphical method is proposed for choosing the time-domain measurements less sensitive to variations in solar irradiance and temperature.

*2.2. Zeta Converter*

The DC-DC converter considered in this paper is a Zeta converter, which is a fourth-order non-inverted step-up/step-down circuit that guarantees high voltage gain and low ripples in the output voltage and current [18,29]. A Simulink model is used to verify the operation of the converter connected to the photovoltaic source and that of the monitoring method.

Figure 2 shows the whole system reproduced in Simulink and used during the simulation procedure: a Pulse Width Modulation (PWM) technique is implemented to drive the converter switches $S_1$ and $S_2$ (N-channel Power MOSFET) with opposite phases.

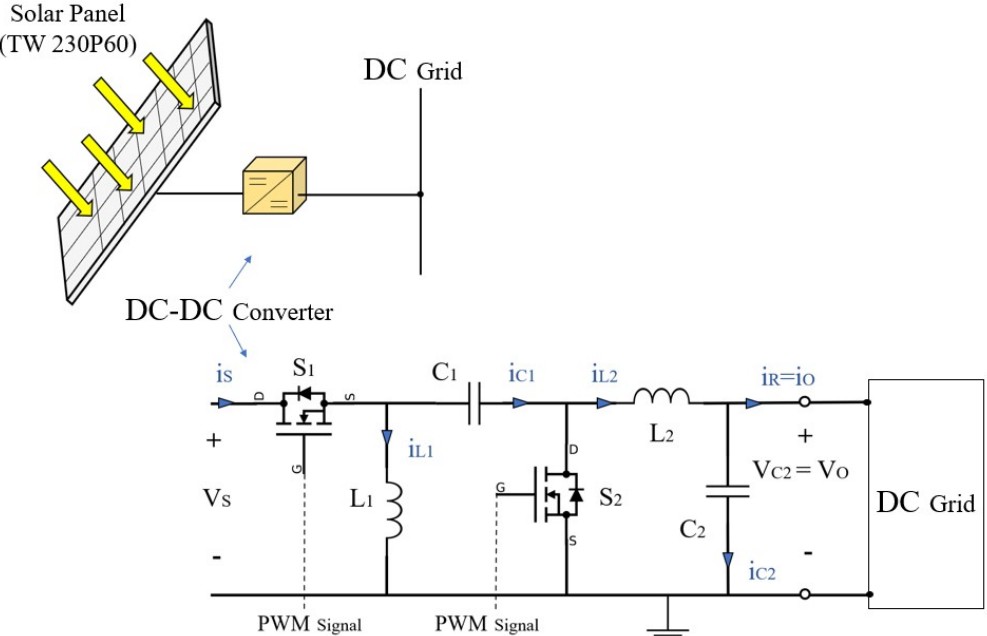

**Figure 2.** General diagram of the photovoltaic system and Zeta converter circuit.

When the switch $S_1$ is in conduction mode, the inductor $L_1$ absorbs energy from the DC source, and, at the same time, $L_2$ absorbs energy from the source and capacitor $C_1$. This means that the input current $i_S(t)$ is equal to the sum $i_{L1}(t) + i_{L2}(t)$, and these two currents increase linearly, as shown in Figure 3a,b, which are extracted from the Simulink model considering a solar irradiance of 1000 W/m$^2$ and an operating temperature of 25 °C. In the opposite condition ($S_1$ Off and $S_2$ On), the input current is zero, and the current $i_{L1}(t)$ flows through $S_2$ to charge capacitor $C_1$. Simultaneously, $i_{L2}(t)$ crosses the circuit ($C_2$-$R$) and returns through the closed switch $S_2$.

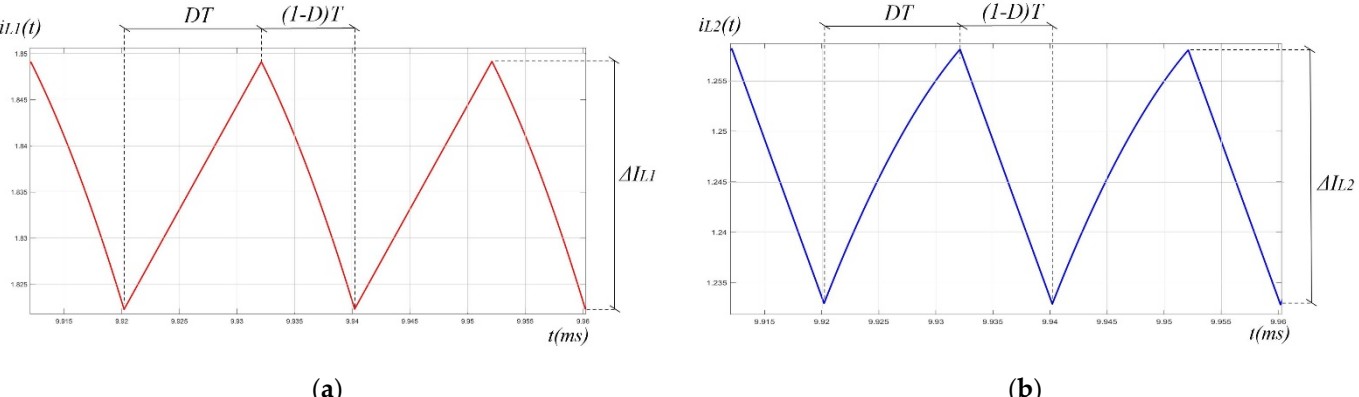

**Figure 3.** Converter currents in time domain: (**a**) current $I_{L1}$; (**b**) current in the inductor $I_{L2}$.

The currents $i_{L1}(t)$, $i_{L2}(t)$, and $i_{S2}(t)$ present three different ripples, $\Delta i_{L1}(t)$, $\Delta i_{L2}(t)$, and $\Delta i_{S2}(t)$, respectively. The latter is the most important to define the conduction mode of the circuit: if current $i_{S2}(t)$ becomes zero during the Switch-Off period, the converter works in the Discontinuous Conduction Mode (DCM). Otherwise, the Continuous Conduction Mode (CCM) requires a non-zero current in $S_2$ when the switching from off to on mode occurs. By operating in CCM, it is possible to reduce the electrical stress on the converter components and obtain a lower ripple on the output quantities. For this reason, only the CCM is considered in this work, and the dimensioning of the analog components has been performed to ensure this condition.

As shown in [18], the voltage gain *G* of the Zeta converter can be calculated through Formula (1), where $V_S$ is the input voltage imposed by the photovoltaic source and $V_O$ is the output voltage of the converter. Consequently, Formula (2) is used to describe the relationship between the mean value of the input current $I_S$ and the output current $I_O$.

$$G = \frac{V_O}{V_S} = \frac{D}{1 - D} \tag{1}$$

$$I_O = \frac{1 - D}{D} I_S \tag{2}$$

As for the dimensioning of the passive components, one of the main aspects is to ensure a sufficient margin between CCM and DCM. In addition, ripples of voltages and currents are also considered, as shown in [18]. The variation ratios of the currents $i_{L1}$ and $i_{L2}$ are expressed by the terms $\eta$ and $\zeta$, respectively, and are calculated as follows,

$$\eta = \frac{\Delta_{i_{L1}}/2}{I_{L1}} = \frac{1 - D}{2\,G} \frac{R}{f\,L_1} \tag{3}$$

$$\zeta = \frac{\Delta_{i_{L2}}/2}{I_{L2}} = \frac{D}{2\,G} \frac{R}{f\,L_2} \tag{4}$$

where capital letters are used to indicate the average values of the quantities, and *R* represents the load resistance. Similarly, the variation ratios of the voltages are calculated as,

$$\rho = \frac{\Delta_{v_C}/2}{V_{C1}} = \frac{D}{2} \frac{1}{f\,C_1\,R} \tag{5}$$

$$\varepsilon = \frac{\Delta_{v_0}/2}{V_o} = \frac{D}{8\,G} \frac{1}{f^2\,C_2\,L_2} \tag{6}$$

where $\rho$ is the variation ratio of the voltage across $C_1$, and $\varepsilon$ is that of the output voltage.

Table 2 summarizes the nominal values of the converter components that ensure CCM operation and limit the output ripple to 5%.

**Table 2.** Converter components.

| $L_1$ [mH] | $L_2$ [mH] | $C_1$ [µF] | $C_2$ [µF] |
|:---:|:---:|:---:|:---:|
| 5 | 5 | 2.4 | 2.4 |

### 2.3. Fault Classes

To propose a prognostic approach for photovoltaic systems focused on parametric faults, the corresponding classes must be defined starting from tolerance ranges around the component nominal values. In fact, parametric faults are deviations of components from the nominal values that produce a partial loss of functionality. These variations could introduce undetectable consequences from a general point of view in the functioning of the system but by choosing appropriate measurements, the variations can be identified and localized in a specific component or in a group of components.

The nominal intervals of the four passive components are defined, starting from the nominal values shown in Table 2 and applying a 15% tolerance. These variations are considered acceptable as they guarantee an output ripple lower than 10% and maintain CCM operation. The parametric failure conditions correspond to a maximum decrease of 70% for each passive component. Table 3 summarizes the operating ranges of each component.

**Table 3.** Operating ranges.

|  | $L_1$ (mH) | $L_2$ (mH) | $C_1$ (µF) | $C_2$ (µF) |
|:---:|:---:|:---:|:---:|:---:|
| Nominal Range | (4.25–5.75) | (4.25–5.75) | (2.04–2.76) | (2.04–2.76) |
| Malfunction Condition | (1.5–4.25) | (1.5–4.25) | (0.72–2.04) | (0.72–2.04) |

It is necessary to highlight that the single failure hypothesis is assumed because it is the most likely, and no-fault propagation is expected. This means that only one passive component at a time can be considered defective, and five classes of failure are used. The nominal working condition of the converter is called "class 0", and it presents all components in their nominal ranges. The other classes are summarized in Table 4.

**Table 4.** Fault Classes.

| Fault Class | Description |
|:---:|:---:|
| 0 | Each component is in the nominal range |
| 1 | Malfunction on $L_1$ |
| 2 | Malfunction on $L_2$ |
| 3 | Malfunction on $C_1$ |
| 4 | Malfunction on $C_2$ |

Therefore, the main objective of the classifier is to identify these working conditions starting from specific measurements extracted from the DC-DC converter. To make the monitoring system suitable from a practical point of view, it is important to offer a low intrusive level using as few measures as possible. For this reason, in the next paragraphs of this paper, a selection method of the measurements is proposed based on the testability level of the circuit and on the influence of the environmental conditions.

All the time-domain measures considered in this work have two information contents: the first is linked to the average value of the quantities and the second to their ripples. Therefore, the dataset matrix used to train the neural classifier must contain two columns for each measurement and one column for the corresponding class. The general form of the dataset is (7),

$$\begin{bmatrix} Q_{1,m}^{1} & Q_{1,r}^{1} & \cdots & 0 \\ Q_{1,m}^{2} & Q_{1,r}^{2} & \cdots & 0 \\ \vdots & \vdots & \cdots & \vdots \\ Q_{1,m}^{N_S} & Q_{1,r}^{N_S} & \cdots & 4 \end{bmatrix} \tag{7}$$

where, for example, $Q_{1,m}^1$ is the first measure of the mean value of the quantities $Q_1$ (voltage or current) and $Q_{1,r}^1$ is its ripple. A significant number of samples are used for each fault class, and the total number of rows belonging to the dataset is $N_S$. It should be noted that keeping the duty cycle fixed; it is possible to reduce the size of the dataset matrix and the variability of the measurements. In fact, the values of the measured quantities in the nominal conditions are known, and the recognition of malfunctions is facilitated. If the duty-cycle is continuously changed, it would be necessary to add a column containing the different values of D and replicate the structure shown in (7) for each value of D.

*2.4. Testability Analysis*

Testability analysis represents a fundamental step in each diagnostic and prognostic system. Thanks to this study, the identifiable components are defined, thus facilitating the selection of test points and the definition of the achievable objectives.

Since the Simulation After Test (SAT) techniques are usually used to detect parametric faults in analog circuits, the main step is the definition of the failure equations. These equations present the component values as unknowns and are obtained by comparing the circuit responses at specific points with their nominal forms. The solvability level of the failure equations corresponds to the testability of the circuit, and its maximum value is the total number of passive components. If the testability is less than the maximum value, one or more ambiguity groups can be defined: they are sets of components that cannot be uniquely determined starting from the selected measurements.

Several methods have been developed in recent years in order to facilitate the calculation of testability in different types of electrical circuits [30]. The study of linear time-invariant circuits is widespread in the literature, and the calculation methods in the frequency domain can be easily adapted to different topologies [31]. Regarding the non-linear time-variant circuits, the testability evaluation presents many complexities due to the different topologies that the circuit assumes during operation. In the case of DC-DC converters, two different topologies are used in Continuous Conduction Mode (CCM), one for each switching period, while in Discontinuous Conduction Mode (DCM), the topologies become three due to the cancellation of the inductor current.

In such cases, a time-domain method can be used [32]. The first step is to choose a specific switching period in steady-state conditions and to sample the inputs, which are the circuit power supply quantities. In this way, vector $\boldsymbol{u_0}$ is obtained. The second step is the definition of the output vector, as shown in (8), where $\boldsymbol{p}$ is the vector of the unknown parameters and $\boldsymbol{y}_T$ is a vector of measurements obtainable from the circuit.

$$\boldsymbol{y}_T(\boldsymbol{p},\boldsymbol{u_0}) = \left[ y(t_1,\boldsymbol{p},\boldsymbol{u_0})^{tr}, y(t_2,\boldsymbol{p},\boldsymbol{u_0})^{tr}, \ldots, y(t_n,\boldsymbol{p},\boldsymbol{u_0})^{tr} \right]^{tr} \tag{8}$$

Then, the failure equations can be obtained by (9),

$$\boldsymbol{y}_T(\boldsymbol{p},\boldsymbol{u_0}) = \boldsymbol{y}_T{}^* \tag{9}$$

where $\boldsymbol{y}_T{}^*$ is the vector of measurements extracted from the circuit. Finally, testability is calculated as the rank of the Jacobian matrix obtained stating from these equations.

The equivalence between the time-domain procedure proposed here and the testability analysis performed in the Laplace domain is presented in [33]. In this paper, the testability assessment of the Zeta converter is carried out through SapWinPE (SapWin for Power Electronics), which is a program for simulating analog switching circuits. A specific algorithm called TAPSLIN (Testability Analysis for Periodically switched Linear Networks) is implemented on SapWinPE to perform the testability analysis in symbolic form [32].

*2.5. Multilayer Neural Network with Multi-Valued Neurons (MLMVN) and Its Adaptation to a Zeta Convertor*

2.5.1. Main Characteristics

The main tool proposed in this paper for identifying the state of health of DC-DC converters is a neural network-based classifier. It is a feed-forward Multilayer neural network with Multi-Valued Neurons that uses a derivative-free learning algorithm during the backpropagation procedure. Each neuron is a Multi-Valued Neuron (MVN) with multiple complex-valued inputs $(X_1, \dots, X_n)$ and complex-valued weights $(W_1, \dots, W_n)$. This neural network might be used either in a continuous or in a discrete version.

A three-layer neural network with discrete output neurons is used in this paper: each neuron belonging to the last layer employs a discrete activation function dividing the complex plane into k equal sectors and adjusting the output to the lower border of the sector that contains the weighted sum of the inputs $z$ $(z = X_1 W_1 + X_2 W_2 + \dots + X_s W_n)$. The discrete activation function is represented by Formula (10), as it is described in [34].

$$P(z) = Y = \varepsilon_k^j = e^{i2\pi j/k} \quad if \ \ 2\pi j/k \leq \arg(z) < 2\pi(j+1)/k \tag{10}$$

where $j$ is an index of a sector where the weighted sum is located, $k$ is the total number of the sectors, and $\arg(z)$ represents the argument of the weighted sum.

Since this neural network is feed-forward, the backpropagation of the output errors is the main procedure for the correction of the complex weights during the training phase. These errors are calculated starting from a dataset containing corrected classification examples and applying the correction rules in a supervised procedure. As it is shown in (3), the last column of the dataset matrix contains the desired outputs, which are the fault classes corresponding to the time-domain measurements. The dataset rows are processed in succession during the training phase, and the error value for each neuron in the output layer is calculated by the difference between the number (a root of unity) determining the lower border of the desired sector and the actual output. Therefore, the error for each sample belonging to the dataset corresponds to the difference between two complex numbers located on the unit circle.

Applying the standard correction rules presented in detail in [34], it is possible to modify the complex weights without introducing derivative terms, thus facilitating the backpropagation procedure compared to other feed-forward neural networks and reducing the computational cost. Formula (11) shows how to calculate the adjustment of a neural network weight,

$$\Delta W_i^{k,m} = \frac{\alpha_{k,m}}{(n_{m-1}+1)\left|z_{k,m}^s\right|} \delta_{k,m}^s \overline{Y}_{i,m-1}^s \tag{11}$$

where $\Delta W_i^{k,m}$ is the adjustment for the $i$-th weight of the $k$-th neuron belonging to the layer $m$, $\alpha_{k,m}$ is the corresponding learning rate (complex-valued in general, but set equal to 1 in all actual applications), $n_{m-1}$ is the number of the neuron inputs equal to the number of the outputs of the previous layer, $\left|z_{k,m}^s\right|$ is the magnitude of the weighted sum, $\delta_{k,m}^s$ is the neuron error obtained through the backpropagation method, and $\overline{Y}_{i,m-1}^s$ is the conjugate-transposed of the input.

This learning rule allows the correction of the weights for each sample of the dataset $s$ $(s = 1, \dots, N_S)$. While the error should be backpropagated starting from the output neurons up to the input ones, after all the neurons errors were found, the weights should be adjusted starting from the first hidden layer to the last one. Figure 4 shows the initial error definition for a neuron in the output layer.

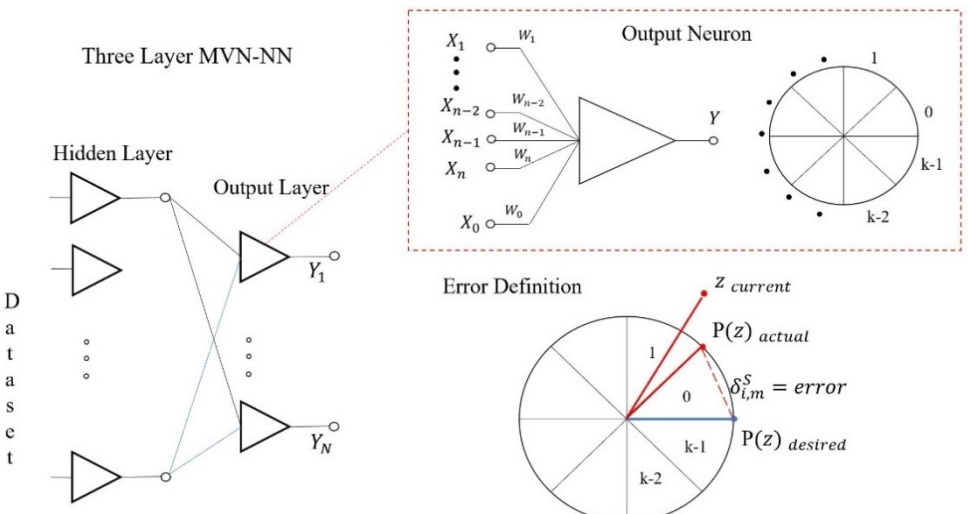

**Figure 4.** General configuration of the MLMVN and error definition in the output layer.

Once the errors have been calculated for the output layer neurons, the errors should be backpropagated by applying Formula (12).

$$\delta_{k,m-1}^{S} = \frac{1}{(n_{m-2}+1)} \sum_{i=1}^{n_m} \delta_{i,m}^{S} \left( W_{k}^{i,m} \right)^{-1} \tag{12}$$

In order to simplify the training procedure reducing the computational cost, a batch algorithm based on the Linear Least Square (LLS) method can be introduced, as shown in [35]. In this case, the errors calculated for each sample belonging to the dataset are saved in a specific matrix without correcting the complex weights. When all samples have been processed, this matrix presents $N_S$ rows, as shown in (13). Since the number of samples is greater than the number of weights, an oversized system of equations is obtained, and different techniques can be applied, such as Q-R decomposition and Singular Value Decomposition (SVD), reducing the number of iterations required for the calculation of the corrections.

$$\begin{bmatrix} \delta_{1,m}^{1} & \delta_{2,m}^{1} & \cdots & \delta_{n,m}^{1} \\ \delta_{1,m}^{2} & \delta_{2,m}^{2} & \cdots & \delta_{n,m}^{2} \\ \vdots & \vdots & \cdots & \vdots \\ \delta_{1,m}^{N_S} & \delta_{1,m}^{N_S} & \cdots & \delta_{n,m}^{N_S} \end{bmatrix} \tag{13}$$

To improve the performance of the classifier, the soft margin technique described in [36] is used. In this case, the purpose of the correction is not only to position the outputs within the desired sectors but, for each of them, to minimize the distance of all output neuron-weighted sums from the bisector of the desired sector as much as possible. In this way, better classification results are obtained avoiding the ambiguity caused by outputs close to the boundary of two different sectors.

### 2.5.2. Neural Classifier for Zeta Converter

In order to adapt the MLMVN to the objective of the paper, a number of binary neurons equal to that of the passive components are used in the output layer. The binary neurons divide the complex plane into two different sectors: the first corresponds to the upper half plane $[0\ \pi)$ and is identified by the value 0; the second sector corresponds to the phase interval $[\pi\ 2\pi)$ and is encoded by the number 1. As it is shown in Figure 5, the first (labeled by 0) sector is used to indicate the nominal behavior of the corresponding component, while the second sector (labeled by 1) is used to describe its malfunction condition.

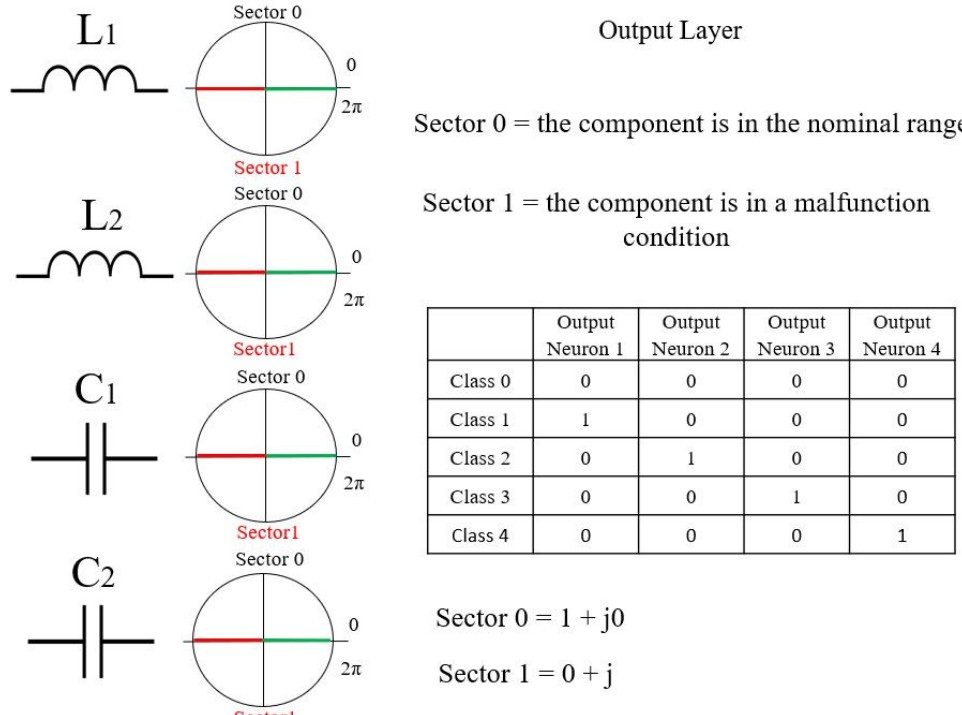

**Figure 5.** Set-up of the output layer and coding of the classes.

When, for example, only the first output neuron is "high", the first fault class is identified, which means that a problem is detected on $L_1$. This rule is used for classes 1 to 4, while class 0 corresponds to a "low" value on each output neuron. It is necessary to highlight that the "winner take all" technique is used to avoid the presence of two outputs on the "high" value at the same time. This means that only the minor error output is considered equal to 1. Formula (14) describes the dataset matrix introducing the coding of the fault classes.

$$
\begin{bmatrix}
Q^1_{1,m} & Q^1_{1,r} & \dots & 0\ 0\ 0\ 0 \\
Q^2_{1,m} & Q^2_{1,r} & \dots & 0\ 0\ 0\ 0 \\
\vdots & \vdots & \dots & \vdots \\
Q^{N_S}_{1,m} & Q^{N_S}_{1,r} & \dots & 0\ 0\ 0\ 1
\end{bmatrix}
\tag{14}
$$

As for the measurements belonging to the dataset, they are used to create complex inputs as follows: each value corresponds to the phase of a complex number with a magnitude equal to 1. These numbers are the inputs of the MLMVN.

## 3. Results

This paragraph presents the simulation results obtained by applying the prognostic approach to the photovoltaic system described above. The main steps of the simulation procedure can be summarized as follows:

- first selection of measurements;
- testability analysis;
- neural network training.

### 3.1. First Selection of the Measurements

The first step of the prognostic procedure is the selection of the most significant measurements to obtain a correct evaluation of the converter status. Since the DC source corresponds to a photovoltaic panel, the input voltage and current are highly variable and depend on the incident solar irradiance and the temperature of the panel. This means that variations in measured quantities can be introduced due to changes in environmental

conditions. If these measurements are used as inputs to the monitoring system, classification errors may occur. To avoid this issue, measurements on the component's quantities are first normalized against input quantities (which are in general known due to their use in almost any MPPT algorithm). Among the normalized quantities, the ones with lower sensitivity towards irradiance and temperature are chosen as inputs for the prognostic classification.

Therefore, the best choice of measurements includes all voltages and currents with low sensitivity compared to the input ones.

As said before, the dataset used to train the neural network-based classifier contains measurements of ripples and mean values. Initially, all currents and voltages on the passive components are taken into consideration, and their variations with respect to the irradiance and temperature are graphically evaluated through the Simulink model. The operating points considered for this simulation are extracted from [37] and represent common situations with a realistic relationship between irradiance and working temperature. Table 5 summarizes these working points.

**Table 5.** Operating conditions.

| Operating Point | Irradiance (W/m$^2$) | Temperature (°C) |
| :---: | :---: | :---: |
| A | 400 | 15 |
| B | 800 | 45 |
| C | 1200 | 65 |

Starting from a common situation characterized by an irradiance of 1000 W/m$^2$ and a working temperature of 55 °C, and considering the fixed grid voltage of 48 V, the maximum power point is obtained with a duty cycle of 0.6. This working condition is chosen as the starting point to evaluate the effects of changes in environmental conditions. Therefore, the duty-cycle is set at 0.6, and the changes in voltages and currents across the passive components are analyzed by moving to the three operating conditions presented in Table 5. Since the environmental situation changes but the duty-cycle is kept constant, the three working points shown in Figure 6 are obtained. These three points indicate three different pairs of input voltage and current. As previously said, to correctly choose the quantities to be measured during the monitoring procedure, the sensitivity of all voltages and currents with respect to these changes is studied.

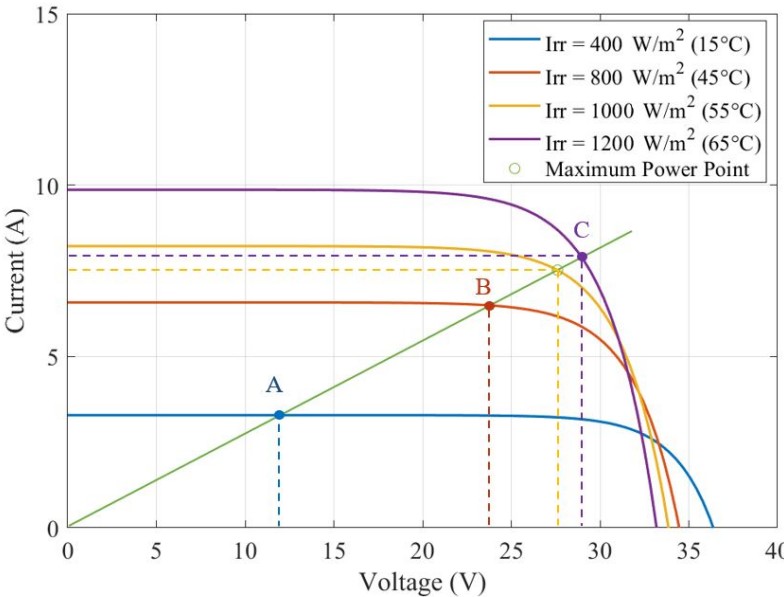

**Figure 6.** Variations of the working point and variations of the input current and voltage.

It should be noted that the duty-cycle value used in this paper is not mandatory and that several methods can be used to choose it. In this case, the starting point is a condition of maximum power transfer, and this value of D is maintained. Once this parameter has been chosen, it is necessary to keep it constant during the generation of all the samples belonging to the dataset matrix.

In this paper, the sensitivity analysis is performed graphically by using the Simulink model described above.

Analyzing the simulation results, it can be observed that the voltage ripples on the passive components and the average values of the currents exhibit a low level of sensitivity with respect to the irradiance and temperature variations. For this reason, the average values of the inductor currents and the ripples of the capacitor voltages are selected as possible measurements. Figure 7a–d presents the approximately constant behavior of these quantities with respect to the changes in input current and voltage.

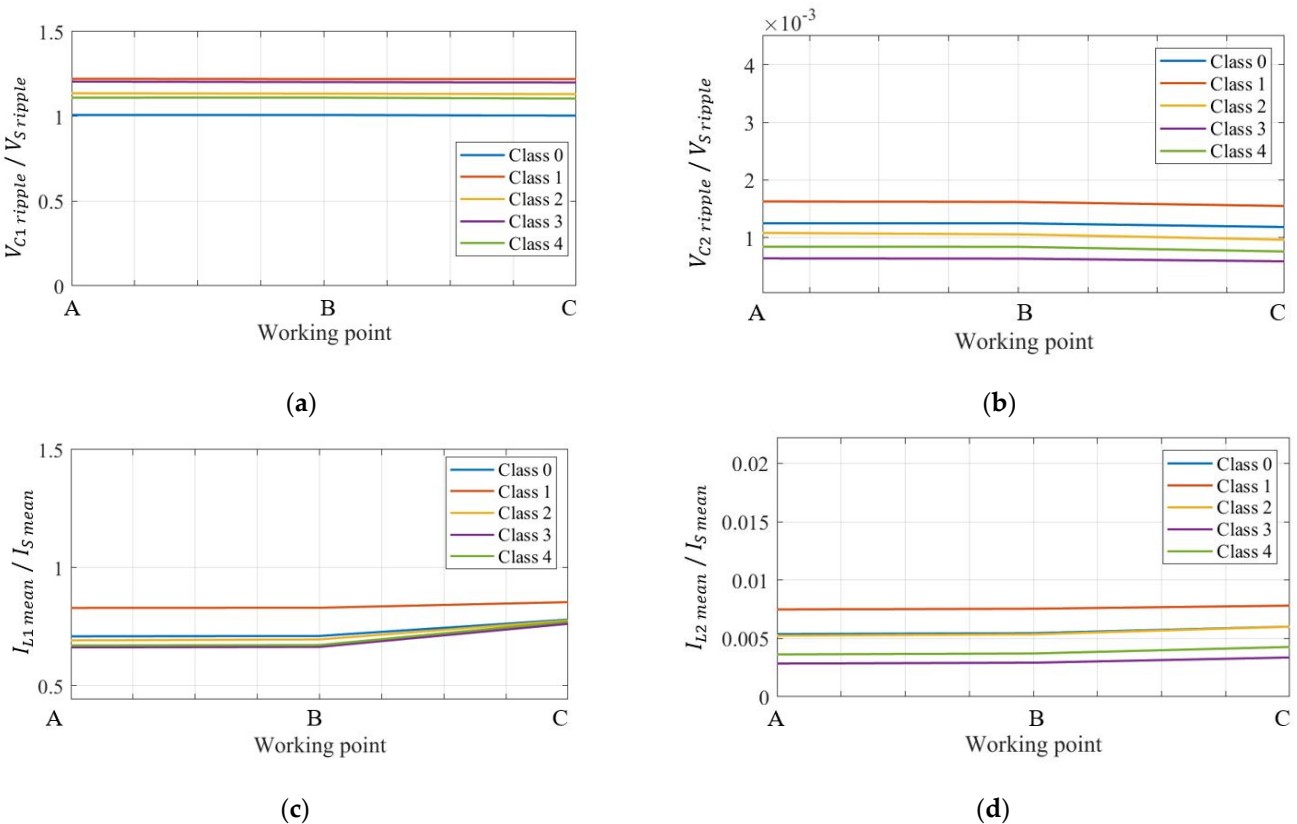

**Figure 7.** Sensitivity of the measurements with respect to the variation of environmental conditions; (**a**) ripple of the voltage on the first capacitor $V_{C1ripple}$; (**b**) ripple of the voltage on the second capacitor $V_{C2ripple}$; (**c**) mean value of the current through the first inductor $I_{L1mean}$; (**d**) mean value of the current through the second inductor $I_{L2mean}$.

### 3.2. Testability Assessment of the Zeta Converter

The testability analysis of the Zeta converter is performed following the theoretical approach described above and using the software TAPSLIN. Figure 8 shows the symbolic circuit developed on SapWin and the consequent analysis in the Laplace domain. The test points used in this case are those corresponding to the previously selected measurements. Therefore, the voltages across the capacitors and currents flow through the inductors are considered for the testability evaluation.

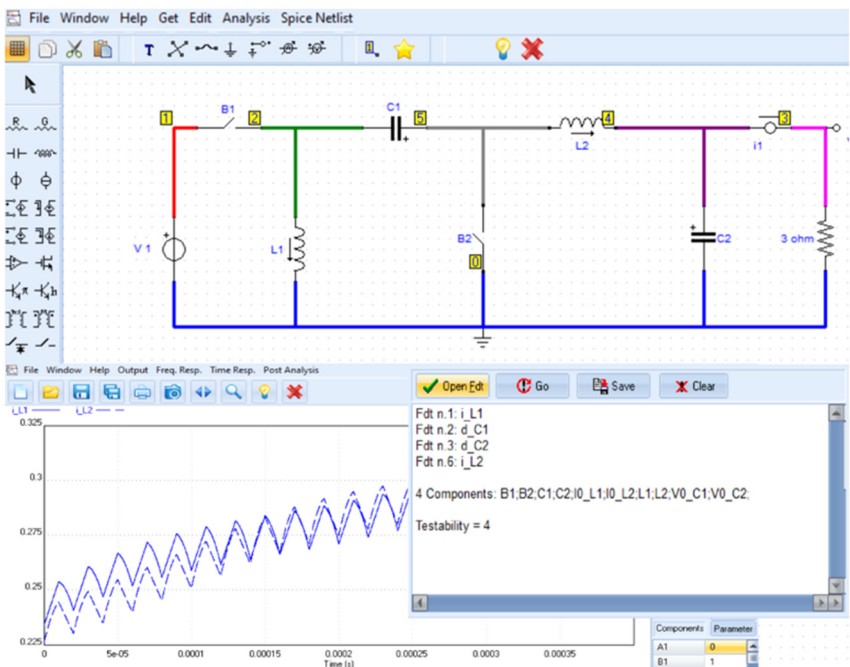

**Figure 8.** Testability analysis of the Zeta converter through SapWin and TAPSLIN.

The results obtained show the absence of ambiguity groups and guarantee the possibility of detecting malfunctions in each passive component. Therefore, the dataset used to train the classifier contains the measurements of two voltage ripples and two current average values (15).

$$
\begin{bmatrix}
V_{C1\;r}^{1} & V_{C2\;r}^{1} & I_{L1\;m}^{1} & I_{L2\;m}^{1} & 0\;0\;0\;0 \\
& & \vdots & & \\
V_{C1\;r}^{N_S} & V_{C2\;r}^{N_S} & I_{L1\;m}^{N_S} & I_{L2\;m}^{N_S} & 0\;0\;0\;1
\end{bmatrix}
\tag{15}
$$

### 3.3. Neural Network Training and Validation

The training of the MLMVN is performed through a Matlab application developed by the authors. This algorithm processes the dataset matrix (11), modifying the complex weights through a Q-R decomposition. The simulation procedure used to create the dataset can be summarized as follows:

- the first step is the creation of 400 random values in the nominal range and 100 random values in the malfunction condition for each passive component;
- using these values, 100 samples for each fault class can be obtained in the hypothesis of a single failure;
- the values of the components are used in Simulink to simulate different working conditions and extract the corresponding measurements (voltage ripple on capacitors and mean current values on inductors);
- repeating these steps for three irradiance values (400, 800, and 1200 $W/m^2$), a dataset matrix containing 1500 samples is obtained.

The three environmental conditions used to create the dataset matrix allow the covering of an extremely wide range of possible scenarios. In this way, it is possible to train MLMVN in a very short time and to exploit its generalization capability to correctly classify many operating conditions not present during the learning phase. Once the dataset has been created, the cross-validation method is used to perform the training of the MLMVN. This means that two phases are performed: the first, called the learning phase, uses 80% of the samples belonging to the dataset for the correction of the weights, while in the second step, called the test phase, the performance of the classifier is verified using the remaining 20% of the samples. The same data split is maintained for complete training, and then it is

changed five times to use all samples both in the learning and test phases. Whenever the data for training and testing are changed, the weights are initialized to random values.

Figure 9a shows the global classification results obtained during the training procedure, while in Figure 9b, the performance of the classifier for each fault class is presented in a histogram chart. In both cases, the index used to evaluate the accuracy of the MLMVN is the Classification Rate (CR), defined as the ratio between the number of correctly classified data and the total number of processed data.

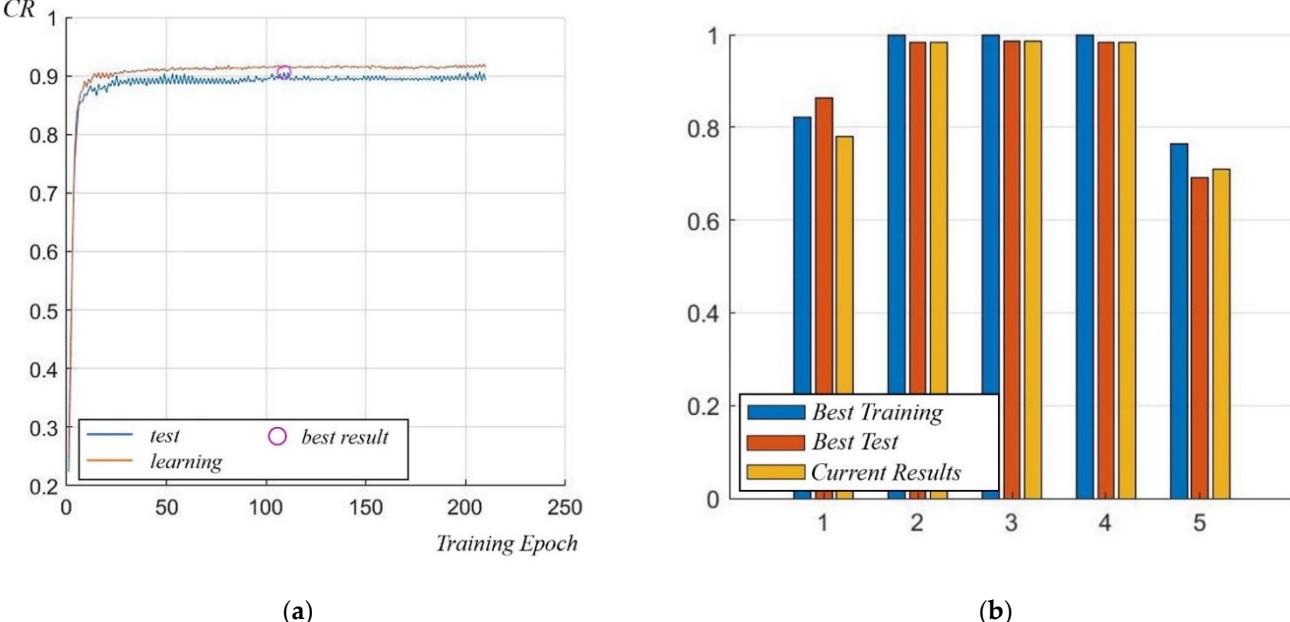

(a)

(b)

**Figure 9.** Classification Results; (**a**) performance of the classifier during the training phase: the red line represents the CR of the learning phase, and the blue line is the CR obtained in the test phase; (**b**) classification results for each fault class shown in the Matlab application at the end of each training epoch.

In order to compare the performance of the MLMVN with that of the other machine learning techniques, a quadratic SVM algorithm is considered. During the training phase, the SVM presents a classification rate of 88.7%. This result has been obtained by processing the same dataset used for the MLMVN-based classifier and using a cross-validation method. Since the one-against-one method is used during the training phase of the SVM algorithm, 10 binary classifiers are defined, each of which presents 13 support vectors.

As shown in Figure 10a,b, further validations of the results can be achieved using these two classifiers for processing new measurements extracted directly from the Simulink model. Two validations are proposed in this paper: the first is obtained by processing new measurements under the same conditions of the training phase (Figure 10a), while the second uses different values of irradiance and temperature (Figure 10b). In particular, the results shown in Figure 10b have been obtained by randomly setting the four fault conditions in some of the environmental situations shown in Table 6. These operational situations represent some typical values of environmental conditions systems in Italy.

**Table 6.** Real working conditions used for validation.

| Irradiance 1 W/m$^2$ | Temperature 1 °C | Irradiance 2 W/m$^2$ | Temperature 2 °C | Irradiance 3 W/m$^2$ | Temperature 3 °C |
|---|---|---|---|---|---|
| 500 | 25 | 705 | 40 | 390 | 19 |

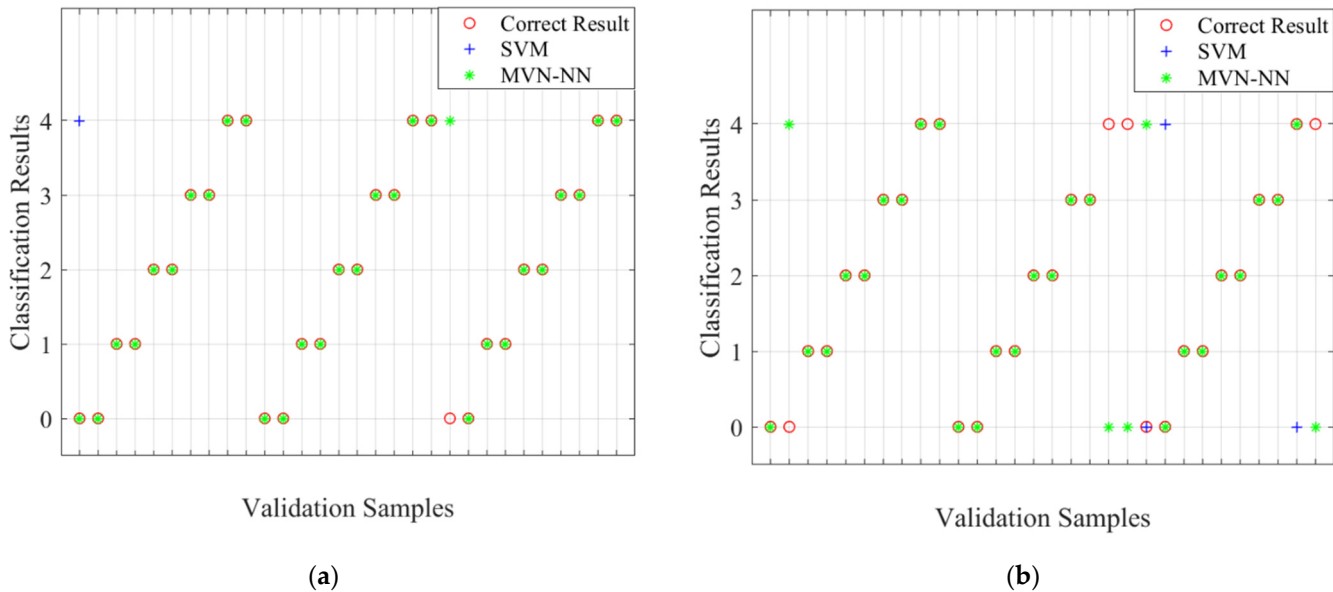

**Figure 10.** Comparison between MLMVN and SVM; (**a**) performance obtained by processing measurements under the same conditions of the training phase; (**b**) performance obtained by processing measurements with different irradiance and temperature values.

Finally, Table 7 summarizes the results obtained during the simulation procedure.

**Table 7.** Simulation Results.

| Classifier | Hyperparameters | Learning Phase | Test Phase | Validation 1 | Validation 2 |
|---|---|---|---|---|---|
| MLMVN | 75 Neurons | 92% | 91.66% | 96.66% | 86.66% |
| SVM | 13 Support Vectors | 88.7% | - | 93.33% | 83.33% |

## 4. Discussion

The results reported in the previous paragraph show excellent performances of the MLMVN-based classifier both in training and in validation.

During the training procedure, the neural classifier with 75 neurons in the hidden layer allows a classification rate of 92% in the learning phase and 91.66% in the test phase. Comparable results can be obtained by increasing the number of neurons in the hidden layer, but this produces a greater difference between the two phases. Therefore, the generalization capability of the neural network decreases by using more than 75 neurons, which means that the CR, obtained by processing new measurements during validation, decreases. Figure 11 summarizes the heuristic procedure used to select the best number of neurons.

As regards the validations, it can be stated that the MLMVN confirms a classification rate higher than 90% using the same conditions as the training, while in the second validation, there is a reduction of up to 86.66%. These results show the possibility of obtaining good performances even without introducing numerous environmental conditions into the dataset used during the training phase.

However, one consideration is needed: observing the results obtained for each class of failure, it can be stated that the main problem is to correctly classify the presence of malfunctions on $C_2$. This aspect is not particularly important when the environmental conditions are similar to those used in training but becomes relevant otherwise. In fact, two false negatives are presented in Figure 10b, and this could be a problem for practical applications. Therefore, even if the classification rate does not decrease significantly, it is advisable to use a dataset with various environmental conditions during the training phase.

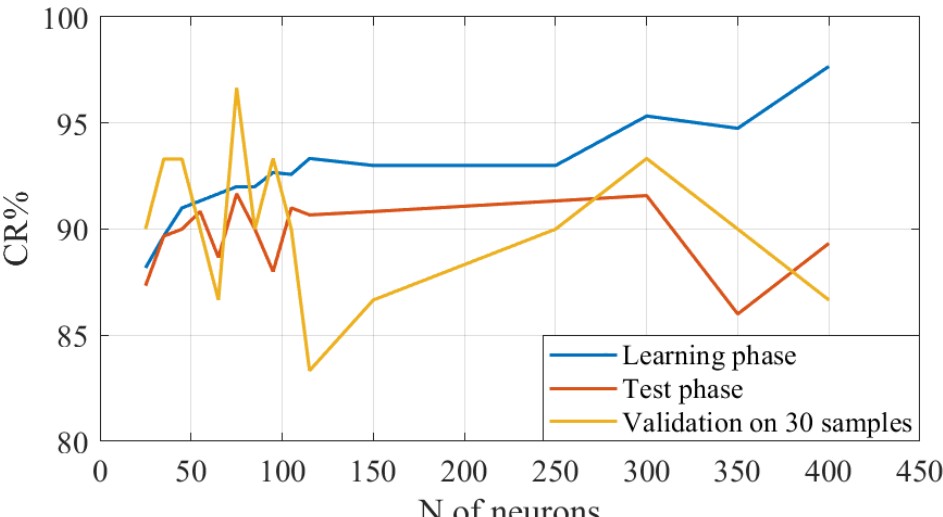

**Figure 11.** Classification rate with respect to the number of neurons in the hidden layer.

## 5. Conclusions

In this paper, a prognostic procedure to monitor the operating conditions of a power converter in photovoltaic applications was proposed. The approach is based on a machine-learning classifier that receives, as inputs, a subset of time-domain measurements of the DC-DC converter and produces, as output, a class that identifies one of four possible faulty components.

To achieve proper fault classification in the presence of an environmental-dependent source, such as a PV device, a normalization procedure was implemented. Among the normalized quantities, a selection of those relevant for testability but insensitive to the irradiance and temperature of the PV source was used. The full system was implemented in Matlab Simulink to generate the datasets used for the classifier training and validation, considering operating conditions compatible with the common operation of a power-producing PV device.

The results from the MLMVN are compared against a standard SVM classifier. The proposed classifier outperforms the SVM both in the training accuracy and in the validation set generalization capabilities.

The simple computational nature of the classifier makes it a prime candidate to be implemented in the embedded environment as well since, differently from deep-learning classification strategies, it shows a very small memory footprint.

**Author Contributions:** Problem identification, I.A.; investigation and conceptualization, F.C. and A.R.; testability analysis and symbolic analysis, M.C.P. and F.G.; neural network application, I.A. and A.L.; procedure development, M.B., G.M.L. and F.C.; simulations, M.B., G.M.L. and F.C. All authors have read and agreed to the published version of the manuscript.

**Funding:** This research received no external funding.

**Institutional Review Board Statement:** Not applicable.

**Informed Consent Statement:** Not applicable.

**Conflicts of Interest:** The authors declare no conflict of interest.

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
