# Peer review of "Machine Learning-Based Monitoring of DC-DC Converters in Photovoltaic Applications"

_algorithms, doi:10.3390/a15030074_

Round 1
Reviewer 1 Report
This manuscript presents a research work, presenting the machine learning based monitoring of DC-DC converters in photovoltaic systems. The scope of research is to prevent catastrophic failures by detecting malfunction conditions on passive components during the operation of Zeta converter. The main technique is based on multi-layer neural network with multi valued neurons. Performance is compared with those of a Support Vector Machine algorithm. Theoretical study is presented with simulation results.
The paper is clearly presented and well-structured but is missing some merit of classifications or boundaries for the implementation and data acquisition.
In conclusion, the paper has its summary by presenting the outcome of a prognostic procedure to monitor small power converter operating conditions in photovoltaic applications. The system was implemented in Matlab Simulink to generate the datasets for training and validation. Results were compared.
General considerations:
- As stated in the Introduction, the primary goal of the research is to prevent catastrophic failures by detecting malfunctioning conditions during regular operation. Why in testing procedures only normal changes in temperature and irradiance were tested? Any dynamic changes in the converter are not considered.
- Why was a zeta converter structure selected as a test converter since it is not the most suitable candidate for photovoltaic energy harvesting due to discontinued input current?
- Why were results of neural network classification not presented on some real environment scenario converter failures?
- Scenarios of irradiance levels over 1000 W/m2 are not applicable on the surface.
Other few comments and suggestions:
1. On page 4, the current is(t), and further on Vs is used, which is not presented in a general diagram in Fig. 2.
2. On page 4, Line 159, “In the opposite condition (S1 On and S2 Off) the input current is zero and the current iL1(t) flows through S2 to charge capacitor C1.” text should be corrected as the state of switches do not correspond the current flow.
3. On page 5, Table 2, the text should be corrected regarding units
4. On page 6, Table 3, the text should be corrected regarding units
5. References used should be unified.
Author Response
We would like to thank the reviewer for the very insightful comments and suggestions expressed during the review process of our article. We reviewed the original manuscript by carefully addressing all comments provided by the reviewer. Major changes can be found in RED throughout the updated manuscript. We believe the overall quality of the paper has been significantly improved by the reviewer's comments.
Attached are the point-by-point responses to the reviewer's comments.

Reviewer 2 Report
In the paper a Zeta converter used in a PV applications is analysed, simulated and used for detecting malfunctions conditions of the system. The prognostic procedure is based on machine learning techniques and focuses on the variations of passive components with respect to their nominal range.
The article is well written and easy to understand, small clarification are needed in some places.
Please specify if the Zeta converter is simulated with ideal components or real?
In real life you have the influence of the PV. You test without normalization? Influenced by the irradiance and temperature?
In the converter design the authors shows the specifications, but, as a reader and engineer interested in replicating some interesting proposal, I only could copy and use the same components, but I can't design a new one for my application. This point is very important because the impact of your work is based on the above: readers understanding your work, drawing inspiration from your work, replicating and improving it. Please add the formulae for dimensioning the passive components, even for Zeta are well-known.
Please add at the references the datasheet of the PV panel.
Please correct the spelling mistakes and add the figures and description of the figures on the same page.
Author Response

(The authors gave the same response as above.)

Reviewer 3 Report
The manuscript presents research related to ensuring the performance of electronic power devices in decentralized power generation systems based on photovoltaic generators. The results were obtained by using techniques of artificial intelligence - machine learning. There are the following remarks and recommendations to the manuscript and its authors:
- editorial - the same expressions and phrases are often repeated, such as: "Therefore, the main purpose of the proposed monitoring method is to detect" pp.63;
- what is the operating frequency of the DC-DC converter and how are the values of the circuit elements determined?
- Is expression 10 written correctly?
- no sensitivity analysis. It is unlikely that changes in the various circuit elements affect the operating mode equally. There are also no recommendations on design and operation. Without this connection, research could be seen as an end in itself;
- the tests were performed in static mode without taking into account the dynamics for three fixed values of the duty cycle (D), and this is quite different from the actual operating mode of a system powered by a photovoltaic generator.
Author Response

(The authors gave the same response as above.)

Round 2
Reviewer 1 Report
The authors did their work. All reviewer's comments have been addressed by the authors and inserted in the text. The paper can be accepted for publication.Reviewer 2 Report
Dear authors,
Congratulations for your work and for the improve that you bring to the paper!
Some small mistakes that you need to correct. In table 2 and 3 appear two times C1. I guess that you wanted to write C1 and C2.
At line 208, it is missed a break between a and 15% (it is written a15%).
Reviewer 3 Report
The authors have comprehensively answered my questions and revised the manuscript according to the recommendations. I have no further comments.